# Nutrition and Sensory Evaluation of Solid-State Fermented Brown Rice Based on Cluster and Principal Component Analysis

**DOI:** 10.3390/foods11111560

**Published:** 2022-05-25

**Authors:** Duqin Zhang, Yanjun Ye, Luyao Wang, Bin Tan

**Affiliations:** Institute of Cereal and Oil Science and Technology, Academy of National Food and Strategic Reserves Administration, Beijing 100037, China; zhduqin@163.com (D.Z.); yeyanjun1997@163.com (Y.Y.); wang-lly@foxmail.com (L.W.)

**Keywords:** brown rice, solid-state fermentation, cluster analysis, principal component analysis, quality evaluation

## Abstract

Consumption of brown rice (BR) contributes to the implementation of the grain-saving policy and improvement of residents’ nutrient status. However, the undesirable cooking properties, poor palatability, and presence of anti-nutritional factors limit the demand of BR products. To enhance its quality, BR was solid-state fermented with single and mixed strains of *Lb. plantarum*, *S. cerevisiae*, *R. oryzae*, *A. oryzae,* and *N. sitophila*. Effects of solid-state fermentation (SSF) with different strains on the nutrition and sensory characteristics of BR were analyzed by spectroscopic method, chromatography, and sensory assessment. Contents of arabinoxylans, β-glucan, γ-oryzanol, phenolic, and flavonoid were significantly increased by 41.61%, 136.02%, 30.51%, 106.90%, and 65.08% after SSF, respectively (*p* < 0.05), while the insoluble dietary fiber and phytic acid contents reduced by 42.69% and 55.92%. The brightness and sensory score of BR significantly improved after SSF. Furthermore, cluster analysis (CA) and principal component analysis (PCA) were employed to evaluate BR quality. Three clusters were obtained according to CA, including BR fermented for 30 h and 48 h, BR fermented for 12 h, and the control group. Based on PCA, the best SSF processing technology was BR fermented with *Lb. plantarum* (0.5%, *v*/*w*) and *S. cerevisiae* (0.5%, *v*/*w*) at 28 °C for 48 h (liquid-to-solid ratio 3:10).

## 1. Introduction

Brown rice (BR) is considered as one of the most important whole grains, which is composed of pericarp, aleurone layer, seed coat, germ, and starchy endosperm [1,2]. As compared to the milled rice (composed of starchy endosperm), the development of BR products achieve a maximized use of grain resource [3]. Moreover, numerous studies have proven that BR contains a variety of nutritional and bio-functional components, including dietary fibers, γ-oryzanol, phenolic acids, and flavonoids [4]. Therefore, the promotion of BR consumption could not only contribute to the grain saving but also improve the residents’ nutrients intake.

However, in reality, the widespread consumption of BR is less than milled rice due to the undesirable cooking properties, poor palatability, and the presence of anti-nutritional factors (such as phytic acid) [5]. Accordingly, many processing approaches (usually classified as physical, biological, and the combination of the both) have been developed to improve quality attributes of BR during the latest years [6,7]. The biological processing mainly involved in exogenous enzyme addition, germination, and biological fermentation was proven to be one of the most effective approaches in improving BR attributes. For instance, the addition of exogenous enzymes such as transglutaminase, amylase, and a couple of cell-wall-degrading enzymes improved the starch digestibility, phenolic content, and antioxidant activity of BR [8,9]. In particular, a great deal of studies have reported that germination could improve the texture of BR, enhance the bio-accessibility of the nutrients (including mineral elements, amino acids, antioxidants, and starch, etc.), and induce the formation of numerous bioactive compounds (such as soluble dietary fiber, gamma-aminobutyric acid, and polyphenols, etc.) [10,11,12].

Fermentation is defined as the course of biochemical modification and refinement of BR (as substrates) by microorganisms and their metabolites [13]. Microbial strains of *Lactiplantibacillus plantarum* (*Lb. plantarum*), *Saccharomyces cerevisiae* (*S. cerevisiae*), and the filamentous fungi (*Rhizopus*, *Aspergillus*, and *Neurospora* etc.) have been used in the fermentation of BR for long time [14,15]. As compared to the exogenous enzyme addition and germination, fermentation with different microbial strains could not only increase the nutritional and bioactive attributes and improve the texture but also enhance the flavor and taste as well as suppress the microbe-dependent spoilage and prolong the shelf life of the BR products [16,17,18]. In addition to the abovementioned merits, adoption of the solid-state fermentation (SSF) could induce higher fermentation productivity, lower catabolic repression, and reduction of production costs [19]. Therefore, SSF tends to be widely applied in the development of BR-based products, including staple food (rice, bread, noodle, etc.), functional foods (probiotic complex), snacks (cake, biscuits, etc.), and beverages [20]. Although the nutritional and function improvement of BR-based products through SSF have been widely studied, most of the inoculations were conducted by only single strains, and the different effects of SSF with various strains are rarely considered.

Our previous pilot study showed that strains of *Lb. plantarum*, *S. cerevisiae*, *R. oryzae*, *A. oryzae*, and *N. sitophila* were the dominant floras in the leavens/starters of BR-based products (bread, cake, pudding, etc.). However, limited information is currently available on the influence of SSF with these different strains on the nutrition and sensory qualities of BR. Therefore, to figure this out, *Lb. plantarum*, *S. cerevisiae*, *R. oryzae*, *A. oryzae*, and *N. sitophila* were single and mixed vaccinated to BR. The nutrition characteristics, including dietary fiber, arabinoxylans, β-glucan, phytic acid, γ-oryzanol, phenolic acid, and flavonoid contents, and the sensory characteristics, including brightness and sensory score, were analyzed. Furthermore, to choose the best microbial strains for the SSF of BR, cluster analysis (CA) and principal components analysis (PCA) were conducted to evaluate the quality of BR according to the nutrition and sensory characteristics. This study could provide basic data and theoretical support for the application of SSF in the processing of BR as well as the development of the new BR-based products.

## 2. Materials and Methods

### 2.1. Materials

The brown rice (BR) cultivar, Suijing No. 18, was kindly provided by Wuchang Jin He Rice Industry Co., Ltd. (Heilongjiang, China). Active dry *Lactiplantibacillus plantarum* (*L**b. plantarum*, CICC 22696), *Saccharomyces cerevisiae* (*S. cerevisiae*, CICC 1223), *Rhizopus oryzae* (*R. oryzae*, CICC 40282), *Aspergillus oryzae* (*A. oryzae*, CICC 41737), and *Neurospora sitophila* (*N. sitophila*, CICC 40204) used in this study were purchased from the China Center of Industrial Culture Collection (CICC; Beijing, China). Yeast peptone dextrose (YPD), de Man, Rogosa and Sharpe (MRS), and potato dextrose broths (PDB), respectively, used as culture medium for *S. cerevisiae*, *L**b. plantarum*, and *R. oryzae*, were commercially available and purchased from AOBOX Reagent Co., Ltd. (Beijing, China). Czapek dox broth (CDB), used as culture medium both for *A. oryzae* and *N. sitophila,* was purchased from Coolaber Reagent Co, Ltd. (Beijing, China). Xylose standards were from Solarbio Science & Technology (Beijing, China). β-glucan and phytic acid (phytate)/total phosphorus assay kits were from Megazyme (Wicklow, Ireland. Chromatographic methanol, Folin–Ciocalteu reagent, gallic acid, and catechins were purchased from Sigma-Aldrich (St. Louis, MO, USA). Antioxidant capacity assay kits (T-AOC) were provided by Nanjing Jiancheng Bioengineering Institute (Nanjing, China). All other chemicals were of analytical grade and purchased from Sinopharm Chemical Reagent (Beijing, China).

### 2.2. Proximate Composition of BR

Moisture content was determined by drying BR samples in an oven at 105 °C overnight to constant weight (Association of Official Agricultural Chemists (AOAC), 925.09). Protein content was determined by the high-temperature combustion Dumas method using an Elementar rapid N cube (Hanau, Germany) and a 5.83 nitrogen-to-protein conversion factor (AOAC 992.15). Fat was determined by the Soxhlet extraction method (AOAC 960.36), ash content was determined by carbonizing the sample in a muffle furnace for 8 h at 550 °C (AOAC, 923.03), and starch content was determined using a total starch assay kit (Megazyme, Wicklow, Ireland).

The proximate compositions of BR (dry basis) were 2.12 g/100 g fat, 1.35 g/100 g ash, 8.60 g/100 g protein, 77.39 g/100 g starch, and 3.65 g/100 g dietary fiber.

### 2.3. Solid-State Fermentation

#### 2.3.1. Pretreatment of BR

The BR was cleaned and soaked (25 °C, 3 h) in potable water to remove impurities, drained, and sterilized at 121 °C for 20 min under high pressure before use.

#### 2.3.2. Activation of Strains

The beakers, shake flasks, centrifuge tubes, pipette tips, and all the other containers were sterilized at 121 °C for 20 min at autoclave (HVA-110; Hirayama Manufacturing Co., Ltd., Tokyo, Japan) before use. A total of 0.1 g of active dry *S. cerevisiae*, *L**b. plantarum*, *R. oryzae*, *A. oryzae,* and *N. sitophila* was, respectively, reactivated in 50 mL YPD, MRS, PDB, and CDB culture medium at 28 °C for 18 h. Then, the four media were centrifuged at 3,000× *g* for 10 min. the pellet was washed with 0.85% physiological saline solution twice and suspended in 50 mL of sterile distilled water. In order to assure the consistency of the inoculation concentration, according to the cell viability result, the *S. cerevisiae* and *L**b. plantarum* suspensions were finally diluted to 10^7^ cfu/mL, while the spore suspensions of *R. oryzae*, *A. oryzae,* and *N. sitophila* were around 10^7^ spores/mL. All the above-mentioned strain suspensions were temporarily stored at 4 °C and used for inoculation within 12 h.

The cell viabilities of *S. cerevisiae*, *L**b. plantarum*, *R. oryzae*, *A. oryzae,* and *N. sitophila* suspensions were determined by spreading the dilutions prepared with 0.85% physiological saline solution on YPD, MRS, PDB, and CDB agar plates, respectively. The growth state of the five strains is shown in Appendix A.

#### 2.3.3. Inoculation

Solid-state fermentation (SSF) of autoclaved BR was performed in sterile, food-grade, transparent plastic bags with one-way air outlet at different *S. cerevisiae*, *L**b. plantarum*, *R. oryzae*, *A. oryzae,* and *N. sitophila* inoculation ratios. Table 1 shows the fermentation conditions for the control and study groups and the labels of the different samples. In order to clearly introduce the result, Arabic numbers of 1, 2, and 3 were affixed after the sample labels to indicate the BR samples fermented for 12 h, 30 h, and 48 h, respectively.

In the clean bench, specific volumes of the five strain suspensions and sterile water (based on Table 1) were added into BR in the sterile, food-grade, transparent plastic bags; sealed; and shaken manually to ensure the BR mixed well with the strain suspensions. All the test groups were statically incubated at 28 °C for 12, 30, and 48 h, respectively; sterilized at 121 °C for 15 min once after SSF; and freeze dried for 72 h. Portions of every groups of BR samples was finely ground and sifted through a 100-mesh sieve, which was used for the determination of the nutrition characteristics and color analysis. The rest was used for the sensory assessment.

### 2.4. Nutrition Characteristics of BR

#### 2.4.1. Soluble (SDF) and Insoluble Dietary Fiber (IDF) Contents

Soluble (SDF) and insoluble dietary fiber (IDF) contents of the BR samples were determined according to the AOAC official method 991.43. Briefly, heat-stable α-amylase, protease, and amyloglucosidase were added in turn under different incubation conditions to remove starch and protein components. DF fractions were obtained as indigestible residues after the enzymatic digestion; the insoluble residues were isolated by filtration, and soluble fiber was precipitated with ethanol. Dried residues correspond to IDF and SDF, respectively. For corresponding corrections, the residual ashes and protein (as Kjeldahl N × 6.25) in the residues were carried out according to AOAC official method 923.03 and 992.15, respectively.

#### 2.4.2. Total Arabinoxylans Content (TAX)

The colorimetric method based on Bressiani et al. (2021) [21] was used to determine TAX. For extraction, 1 g of BR sample was weighted in 50 mL centrifuge tubes, and 25 mL of distilled water was added. The tubes were vortexed for 10 s, 1 mL aliquots were transferred to new Pyrex tubes, and 1 mL of H_2_O was added to bring the final volume to 2 mL. Next, 10 mL of the freshly prepared extracting solution, which was composed of 110 mL acetic acid, 2 mL hydrochloric acid, 5 mL 20% (*w*/*v*) phloroglucinol, and 1 mL 1.75% (*w*/*v*) glucose, were added. The tubes were placed in a vigorously boiling water bath for 25 min and cooled rapidly in flowing cold water. Quantification of TAX was performed by reading absorbance values at 558 nm. The TAX value was calculated from the xylose standard curve and prepared by diluting different concentrations of xylose in distilled water.

#### 2.4.3. β-Glucan Content

The β-glucan content was determined according to AOAC official method 995.16 using a Mixed-linkage Beta-glucan assay kit. Briefly, 100 mg sample was accurately weighed in the tube and mixed with 4 mL sodium phosphate buffer (20 mM, pH 6.5). After incubation at boiling water bath for 2 min, lichenase (0.2 mL, 10 U) was added and incubated at 50 °C for 1 h. Sodium acetate buffer (5.0 mL, 200 mM, pH 4.0) was added and centrifuged at 1000× *g* for 10 min. Then, 0.1 mL dispense aliquots were carefully and accurately transformed into three clean test tubes. Aliquots of 0.1 mL β-glucosidase (0.2 U) in 50 mM sodium acetate buffer (pH 4.0) were added to two of these tubes, with the third one as the reaction blank. Next, 50 mM acetate buffer (0.1 mL, pH 4.0) was added and incubated at 50 °C for 10 min. The measurement was performed at the absorbance of 510 nm against reagent blank within 1 h.
β-glucan (g/100 mL) = 27 × ΔA × F/W,
where ΔA was the absorbance after β-glucosidase treatment minus reaction blank absorbance; F was the factor for the conversion of absorbance values to μg of glucose; W was the dry weight of the sample analyzed in mg.

#### 2.4.4. Phytic Acid Content

The phytic acid content of the fermented BR samples was determined using a phytic acid (phytate)/total phosphorus assay kit provided by Megazyme (Wicklow, Ireland). To extract phytic acid, 1 g of sample was accurately weighted into a 75 mL glass beaker and mixed with 20 mL of hydrochloric acid (0.66 M). The beaker was then covered with foil and stirred vigorously for a minimum of 3 h at room temperature to obtain the phytic acid extraction of the sample. Then, 1 mL of the extraction was transferred into 1.5 mL microfuge tube and centrifuged at 5000× *g* for 10 min. Next, 0.5 mL of the resulting extract supernatant was immediately transferred into a fresh 1.5 mL microfuge tube and neutralized by addition of 0.5 mL of sodium hydroxide solution (0.75 M). The phytic acid content of the sample extraction was determined at the wavelength of 655 nm.

#### 2.4.5. Total γ-Oryzanol Content (TOC)

Extractions and determinations of γ-oryzanol content and composition were conducted according to Pascual et al. (2013) [22] with modifications. To determine TOC, 3.0 g of BR samples were added to 10 mL methanol, extracted by an ultrasonic wave at ambient temperature with a power of 100% for 60 min, and centrifuged at 3000× *g* for 10 min. The clear supernatants were collected and concentrated by vacuum evaporation at 50 °C. The resulting concentrated solutions were diluted to 6 mL with methanol and filtered through a 0.45 μm filtering membrane for TOC analyses.

A Waters (Midford, MA) E 2695 HPLC system equipped with a Waters XSelect HSS T3 column (250 × 4.6 mm, 5 μm), a Waters 2489 UV detector, and an auto-sampler was used for the γ-oryzanol assay. Mobile phase was acetic acid/acetonitrile/methanol, isocratic at 3:44:53 (*v*/*v*/*v*) at a flow rate of 1.4 mL/min. Detection was accomplished at 325 nm wavelength. During the test, the standard solution of γ-oryzanol was prepared to the following concentration gradients: 10, 50, 100, 150, and 200 μg/mL as standard curve. The TOC was calculated using the standard γ-oryzanol curve and expressed as microgram of γ-oryzanol per gram of sample (DW).

#### 2.4.6. Total Polyphenol Content and Total Antioxidant Capacity (TAC)

Extractions of polyphenol in BR samples were performed according to Liu et al. (2015) [23] with modifications. To extract polyphenols, 2 g of BR samples were vigorously mixed with 40 mL methanol. The mixture was extracted by an ultrasonic wave at 40 °C with a power of 100% for 30 min. The extract was then centrifuged at 3500× *g* for 10 min. The clear supernatants were collected and concentrated by vacuum evaporation at 40 °C. The resulting concentrated BR solutions were diluted to 2 mL with methanol and saved in 4 °C and dark for further determination of phenolic and flavonoid content as well as the TAC.

The phenolic content was determined by the Folin–Ciocalteu method [24] with slight modifications. Briefly, 1.0 mL 0.2 N Folin–Ciocalteu reagents were added to 250 μL of BR polyphenol extracts and 500 uL distilled water, followed by sodium carbonate (2.0 mL, 10% *w*/*v*), after 30 min at 30 °C. The mixture was incubated at 30 °C for 30 more minutes, and then, the absorbance was measured at 760 nm wavelength using a UV-1101 spectrophotometer (Techcomp, Shanghai, China). The blank samples were 80% methanol/1% HCl. Total polyphenol content was expressed as mg gallic acid equivalent (GAE) per 100 g dry weight (mg GAE/100 g DW).

The flavonoid content was determined by the NaNO_2_-AlCl_3_·6H_2_O method described by Liu et al. (2015) [23] with slight modifications. Briefly, 0.15 mL 5% sodium nitrite solution was added to 0.5 mL of BR polyphenol extracts, diluted to 2 mL with ultrapure water, and reacted for 5 min. Thereafter, 0.15 mL 10% aluminum chloride hexahydrate solutions were added to the mixtures and reacted for more 5 min, followed by 1 mL 1 M sodium hydroxide solutions under dark conditions. The mixtures were measured at a wavelength of 415 nm after 15 min reaction. The standard curve was made with catechins as a standard sample, and the total flavonoid content was expressed as mg catechins equivalent (CE) per 100 g dry weight (mg CE/100 g DW). The experiment was conducted in triplicate.

The TAC of the solid-state fermented BR samples was determined using BR polyphenol extracts according to the requirements of the antioxidant capacity assay kits (T-AOC) provided by Nanjing Institute of biological engineering in China [25]. The phenolic samples could deoxidize transformed Fe^3+^ into Fe^2+^, which could form complex compound with the phenanthroline substances during the determination process. The absorbance values at 520 nm were recorded. The TAC value was expressed as μmol/g DW.

### 2.5. Sensory Characteristics of BR

#### 2.5.1. Color Analysis

Color analysis of the BR was performed in CIE L*a*b* system by reflectance method using Color i5 spectrometer (XRite, Grand Rapids, MI, USA) set for the following parameters: measuring geometry d/8, illuminant D65, observer 10°, and slit width 25 mm.

#### 2.5.2. Sensory Assessment

The sensory assessment was conducted after the BR samples were steamed. All the BR samples were separately placed in a steam cooker (Supor Co., Ltd., Hangzhou, Zhejiang, China), covered with a lid, and steamed over boiling water for 30 min under atmospheric pressure. The temperature of the obtained steamed brown rice was maintained at 30 °C for sensory assessment. The sensory assessment environment was kept constant for all sessions, and no outside influences were allowed to interfere with the assessments of the samples. The criteria for sensory scoring are shown in Appendix A. Twelve students (equal numbers of male and female) who majored in food engineering were involved to evaluate smell (20 points), appearance (20 points), taste (20 points), overall acceptability (30 points), and texture (10 points).

### 2.6. Statistical Analysis

Statistical analysis was performed on the nutrition and sensory characteristics data using the analysis of variance analysis (ANOVA) with Duncan’s multiple range tests. A *p*-value of 0.05 or less was considered to be statistically significant. Data were expressed as mean ± standard deviations. The analyses were conducted by SAS software version 9.4 (SAS Institute Inc., Cary, NC, USA).

Cluster analysis (CA) and principal component analysis (PCA) were used for discriminating and forming clusters of the solid-state fermented BR based on the grade of nutrition and sensory characteristics. CA is an unsupervised classification technology that was employed for characterize similarities among samples [26]. PCA is an unsupervised technique that can calculate the principal components having the largest variance in order to reduce the dimensionality in data sets and allow the visualization of clusters [27]. CA and PCA analyses were performed using SAS software version 9.4.

In this study, ten variables composed of IDF (*X1*), TAX (*X2*), β-glucan content (*X3*), phytic acid content (*X4*), TOC (*X5*), TAC (*X6*), phenolic content (*X7*), flavonoid content (*X8*), sensory score (*X9*), and L* (*X10*) were used for the CA and PCA analyses.

## 3. Results and Discussion

### 3.1. Nutrition Characteristics of BR

The SDF and IDF contents of the solid-state fermented BR are shown in (Figure 1A,B). The initial contents of SDF and IDF in BR (CK) were 0.30 g/100 g DW and 3.35 g/100 g DW, respectively. After SSF, the SDF content in BR significantly increased from 0.24 g/100 g DW to 2.78 g/100 g DW (*p* < 0.05). In general, with the extension of the fermentation time from 12 h to 48 h, the BR samples fermented with two mixed strains exhibited higher SDF contents than the three mixed stains as well as the single cultures, especially in PC3 (2.78 g/100 g DW), followed by MC3 and MY3. The IDF content obviously decreased from 4.60 g/100 g DW to 1.92 g/100 g DW, reduced by 58.26% in C2. This could have been ascribed to the fact that the cellulolytic enzymes secreted by microbes during SSF produced hydrolysis and a dissolving effect on the IDF (the main component of the BR cell wall), resulting in the decrease of IDF and increase of SDF [28].

The main functional components of DF, including arabinoxylans and β-glucan, were further analyzed in the BR after SSF as shown in Figure 1C,D. The total content of arabinoxylans (TAX) in the control group (CK) was 2.86 g/100 g DW, which increased to 4.05 g/100 g DW after fermenting with the mixed two strains of *R. oryzae* and *A. oryzae* for 30 h (PC2). In general, the BR samples fermented with mixed strains for 30 h were favorable for the improvement of the arabinoxylans content. The β-glucan content in BR was significantly lower than the arabinoxylans content in BR (*p* < 0.05), which increased from 1.61 g/kg DW to 3.80 g/kg DW by 136.02% after fermenting with *S. cerevisiae* for 48 h (in Y3). The increase of TAX and the content of β-glucan after SSF could be ascribed to the proliferation of the microbial strains improving the level of both intracellular and exocellular matrix, which were mainly composed of β-glucan and arabinoxylans [29].

Changes of the phytic acid content in BR after SSF are shown in Figure 1E. The initial phytic acid content (CK) was 1.52 g/100 g DW. MYP2 showed the lowest phytic acid content of 0.67 g/100 g DW, achieving the highest phytic acid degradation rate of 55.92%. The reason for the degradation of phytic acid in BR after SSF might be ascribed to the phytase and phosphatase secreted by microbial strains hydrolyzed phytic acid directly as well as the acidification during SSF further activated the endogenous and exogenous phytase and phosphatase in BR [30].

The results of TOC in solid-state-fermented BR with different strains are shown in Figure 1F. The initial content of γ-oryzanol in BR (CK) was 155.76 mg/100 g DW, which increased by 3.95~30.51% after SSF. In comparison to the mixed cultures, fermentation with a single strain presented higher TOC, especially in the Y3 (203.28 mg/100 g DW), M2 (201.57 mg/100 g DW), and P3 (200.33 mg/100 g DW) samples. The significant increase of γ-oryzanol could be attributed to the optimal pH, higher production of xylanase, and greater activity of feruloyl esterase during fermentation, which benefits the degradation and biotransformation of the esterified ferulate into γ-oryzanol [31].

Effects of SSF with different strains on the phenolic and flavonoid contents are shown in Figure 1G,H. The initial contents of phenolic and flavonoid in BR (CK) were 55.18 mg GAE/100 g DW and 49.26 mg CE/100 g DW, respectively. BR vaccinated with mixed strains of *Lb. plantarum*/*S. cerevisiae* (MY3) showed the greatest phenolic (114.17 mg GAE/100 g DW) and flavonoid contents (81.32 mg CE/100 g DW), which were 2.07 and 1.65 times of the CK, respectively. Most of the phenolic and flavonoid are concentrated in the bran, aleurone layer, and germ of BR [32]. In this study, the obvious increase of the phenolic and flavonoid levels were mainly attributed to the acidification and enzymatic hydrolysis (especially α-amylase, β-glycosidase, and xylanase) during SSF, contributing to the depolymerize of bound phenolic acid and flavonoid from cellulose, lignin, and proteins [33].

Effects of SSF with different strains on the TAC of BR are shown in Figure 1I. The TAC of BR obviously increased from 3.00 μmol/g DW (in CK) to 6.31 μmol/g DW (in C2), improving by 110.33% after SSF (*p* < 0.05). The increased antioxidant capacities of the fermented grains are mainly ascribed to the increased strain metabolites, such as bioactive polyphenol, monosaccharide, and amino acids [34,35,36].

### 3.2. Sensory Characteristics of BR

Results of sensory analysis are depicted in Figure 2. Changes in the brightness (L*) of BR following SSF are summarized in Figure 2A. Values of a* (around 1.6) and b* (around 12) did not change significantly before and after SSF in comparison to L*. The initial L* value of the BR was 78.07, mainly presented by the cortex wrapped on the outer surface of the BR grain. After SSF, the L* significantly increased (*p* < 0.05), especially in the MC2 sample (85.43). This might be ascribed to the fact that the cortex curled and separated from the grain under the acidification and hydrolysis during fermentation, and the exposition of the endosperm increased the lightness of the BR. Usually, the dense fiber cortex wrapped on the surface of BR prevents the gelatinization of starch in endosperm, resulting in a hard texture and undesirable chewiness [37]. SSF treatment improved the exposure of the endosperm, increasing the migration of water into the internal structure of BR, which contributes to the gelatinization of BR during cooking.

The sensory assessment including the smell, appearance, taste, overall acceptability, and texture of the steamed BR with or without SSF is shown in Figure 2B. After SSF, the sensory score significantly increased from 59.82 (in CK) to 74.22 (the highest in PC2) (*p* < 0.05), indicating the obvious improvement of the comprehensive sensory characteristics. On the one hand, the improvement of the smell was due to the production of unique flavors, which depend mainly on the release of volatile compounds when the BR is fermented with various strains, including alcohols, acids, esters, ketones, and aldehydes [35]. Furthermore, the increase of the free amino acid (e.g., glutamic acid and aspartic acid) and the soluble sugar contributed to the fresh and sweet taste of BR, respectively [16]. On the other hand, the metabolites secreted by various strains resulted in the acidification and hydrolysis of BR, contributing to the softening, peeling, and dissolving of the BR cortex. These changes not only weakened the block of the cortex and contribute to the gelatinization of the starch but also benefitted the emission of the volatile compounds, enhancing the appearance, taste, and texture of the steamed BR.

### 3.3. Cluster Analysis

Cluster analysis provides an insight into the data by dividing the dataset of objects into clusters in which the objects are more similar to each other than to objects in other clusters [38]. The clustering result is presented by a dendrogram as shown in Figure 3. The bigger the distance between clusters (DBC), the greater the difference between clusters. All the BR samples could be divided into two clusters (I and II) when the DBC was the maximum value of 2.2053. Cluster I included all the BR samples fermented for 30 h and 48 h, while cluster II was composed of the BR samples fermented for 12 h and the unfermented one (CK). When the DBC = 1.5416, all the samples could be divided into three clusters (i, ii, and iii). Cluster i was the same as cluster I, while clusters ii and iii were composed of the BR samples fermented for 12 h and the CK, respectively. The clustering result indicated that the solid-state-fermented BR was significantly different from the unfermented one (CK). It should be noted that cluster analysis loses relational information about the variables with each cluster and is unable to assess the characteristics of the samples according to the different variables [39]. Therefore, PCA was further conducted to not only provide more information about the relationships between SSF and nutrition and sensory characteristics but also to make a comparison of all the fermented BR and CK [40].

### 3.4. Principal Components Analysis

PCA is a multivariate analysis technique employed to achieve a reduction in dimensionality, data exploration for finding relationship between objects, an estimation of the correlation structure of the variables, and an investigation of how many components are necessary to explain the greater part of variance with a minimum loss of information [41]. PCA was performed to describe the relationship between the SSF treatment and the nutrition and sensory characteristics of BR. After the statistical analysis of all data as shown in (Table 2), the PCA model retained three principal components (*Z1*, *Z2*, and *Z3*), which gave eigenvalues greater than 1.0 (5.216, 2.145, and 1.001, respectively) and explained 83.62% of the total variability. The component 1 (*Z1*) described 52.16% of the variability in the parameters, which was slightly positive and influenced by flavonoid content (*X8*); the component 2 (*Z2*), which accounted for 21.45% of the variability in the parameters, received the main positive contribution from phytic acid content (*X4*) and total arabinoxylan content (*X2*); the component 3 (*Z3*) described 10.01% of the variability in the parameters, which was negatively influenced by total arabinoxylan content (*X2*). Table 3 shows the eigenvalues of the three principal components. Based on the relationship between principal components and variables, the linear relationship model could be produced as
*Z1*= −0.310 *X1* + 0.166 *X2* + 0.342 *X3* − 0.161 *X4* + 0.281 *X5* + 0.346 *X6* + 0.335 *X7* + 0.421 *X8* + 0.298 *X9* + 0.395 *X10*(1)
*Z2*= −0.132 *X1* + 0.557 *X2* − 0.072 *X3* + 0.72 *X4* − 0.104 *X5* + 0.032 *X6* + 0.287 *X7* − 0.022 *X8* − 0.231 *X9* + 0.02 *X10*(2)
*Z3*= 0.206 *X1* − 0.627 *X2* + 0.259 *X3* + 0.381 *X4* − 0.134 *X5* + 0.364 *X6* + 0.036 *X7* + 0.182 *X8* − 0.366 *X9* + 0.184 *X10*(3)

According to the principal components and its proportion, an equation expressed as *Z* = 5.216 *Z1* + 2.145 *Z2* + 1.001 *Z3* could be given to compare the fermented BRs with CK based on the nutrition and sensory characteristics. Table 4 shows the principal component ranking order of all the BR samples. Better ranking refers to greater quality (nutrition and sensory) of the BR samples. MY3 showed the highest score (2336.08), followed successively by C2 (2209.39), MY2 (2199.72), C3 (2168.51), and MYC3 (2159.13). The BR sample without SSF treatment (CK) scored 1310.84, ranking at 29th among the 34 samples, which was better than the MYC1, S1, P1, MYP1, and C1. This indicated that fermented BR with two mixed strains of *Lb. plantarum*/*S. cerevisiae* for 48 h had the best nutrition and sensory qualities, followed by the single culture with *A. oryzae* for 30 h, mixed culture with *Lb. plantarum*/*S. cerevisiae* for 30 h, single culture with *A. oryzae* for 48 h, and mixed culture with *Lb. plantarum*/*S. cerevisiae*/*A. oryzae* for 48 h. In comparison to the CK and BR samples fermented for 12 h, the extension of the fermentation time (30 h and 48 h) contributed to the improvement of the BR qualities.

## 4. Conclusions

The microbial processing of SSF with single and mixed strains of *Lb. plantarum*, *S. cerevisiae*, *R. oryzae*, *A. oryzae*, and *N. sitophila* were performed in BR. Effects of SSF with different strains on the nutrition and sensory characteristics of BR were determined, and CA and PCA analyses were conducted to evaluate the quality of the BR after SSF. The content of soluble dietary fiber, arabinoxylans, β-glucan, γ-oryzanol, phenolic acid, and flavonoid significantly increased, while the insoluble dietary fiber and phytic acid content obviously decreased (*p* < 0.05). For the sensory characteristics, SSF improved the brightness and sensory score of BR. According to the CA and PCA results, the solid-state-fermented BR samples were verified to achieve better nutrition and sensory characteristics than the CK. Fermented BR with two mixed strains of *Lb. plantar**um*/*S. cerevisiae* for 48 h showed the best quality among all the BRs, and the next best was the single fermentation with *A. oryzae* for 30 h. In conclusion, the application of SSF is effective in improving the quality of BR and BR-based products, including both nutrition and sensory characteristics. In the future, changes of the BR structures (cortex and endosperm of the BR) during SSF, relationships between the changes of nutrients and the growth curve of the strains, as well as the manufacture of BR-based products with solid-state-fermented BR should be investigated to promote the application and progress of SSF in BR processing.

## Figures and Tables

**Figure 1 foods-11-01560-f001:**
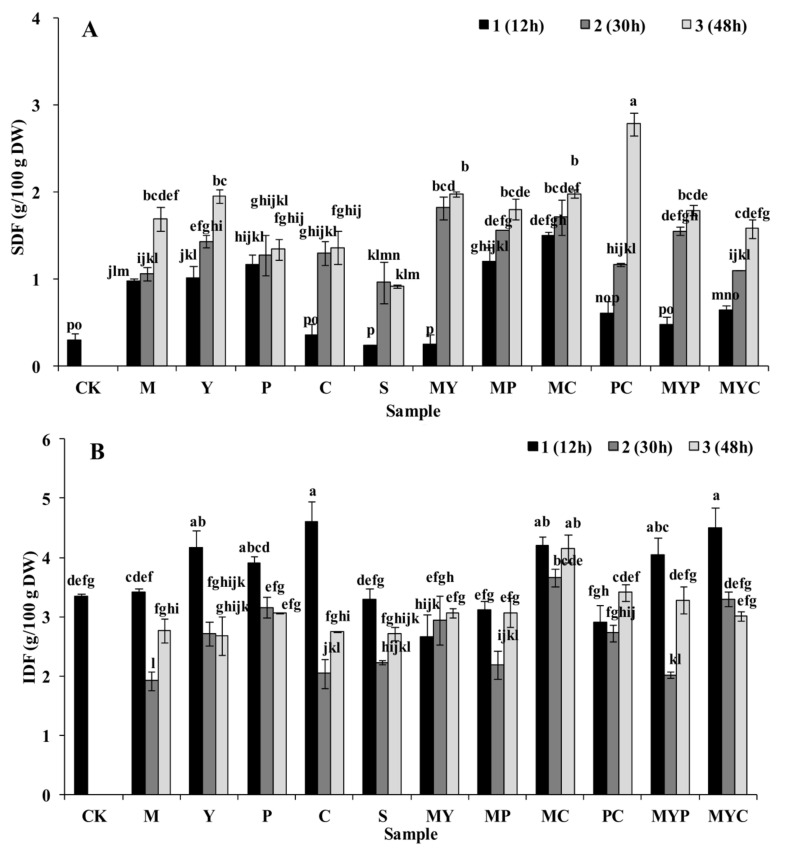
Effect of solid-state fermentation with different strains on the nutrition characteristics of BR. (**A**) Soluble dietary fiber (SDF); (**B**) insoluble dietary fiber (IDF); (**C**) total arabinoxylan content (TAX); (**D**) β-glucan content; (**E**) phytic acid content; (**F**) total γ-oryzanol content (TOC); (**G**) phenolic content; (**H**) flavonoid content; (**I**) total antioxidant capacity (TAC). CK, control group; M, fermented BR with *Lb. plantarum*; Y, fermented BR with *S. cerevisiae*; P, fermented BR with *R. oryzae*; C, fermented BR with *A. oryzae*; S, fermented BR with *N. sitophila*; MY, fermented BR with *Lb. plantarum* and *S. cerevisiae*; MP, fermented BR with *Lb. plantarum* and *R. oryzae*; MC, fermented BR with *Lb. plantarum* and *A. oryzae*; PC, fermented BR with *R. oryzae* and *A. oryzae*; MYP, fermented BR with *Lb. plantarum*, *S. cerevisiae,* and *R. oryzae*; MYC, fermented BR with *Lb. plantarum*, *S. cerevisiae,* and *A. oryzae*. Arabic numbers of 1, 2, and 3 were affixed after the sample labels to indicate the BR samples fermented for 12 h, 30 h, and 48 h, respectively. Data is expressed as the mean ± standard deviation. Different lowercase letters in the columns denote significant difference at the level *p* < 0.05.

**Figure 2 foods-11-01560-f002:**
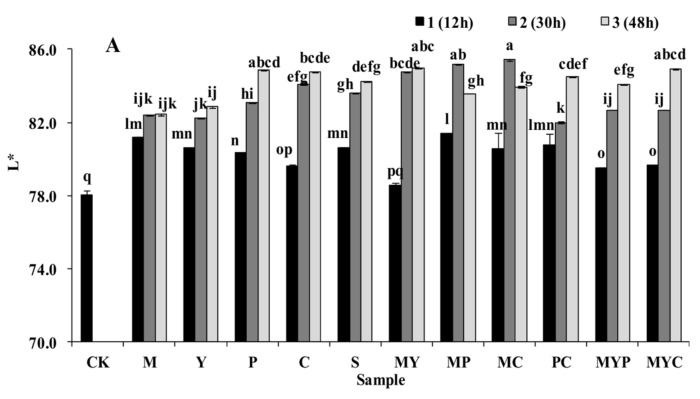
Effect of solid-state fermentation with different strains on the sensory characteristics of BR. (**A**) Brightness (L*); (**B**) sensory score. CK, control group; M, fermented BR with *Lb. plantarum*; Y, fermented BR with *S. cerevisiae*; P, fermented BR with *R. oryzae*; C, fermented BR with *A. oryzae*; S, fermented BR with *N. sitophila*; MY, fermented BR with *Lb. plantarum* and *S. cerevisiae*; MP, fermented BR with *Lb. plantarum* and *R. oryzae*; MC, fermented BR with *Lb. plantarum* and *A. oryzae*; PC, fermented BR with *R. oryzae* and *A. oryzae*; MYP, fermented BR with *Lb. plantarum*, *S. cerevisiae,* and *R. oryzae*; MYC, fermented BR with *Lb. plantarum*, *S. cerevisiae,* and *A. oryzae*. Arabic numbers of 1, 2, and 3 were affixed after the sample labels to indicate the BR samples fermented for 12 h, 30 h, and 48 h, respectively. Data is expressed as the mean ± standard deviation. Different lowercase letters in the columns denote significant difference at the level *p* < 0.05.

**Figure 3 foods-11-01560-f003:**
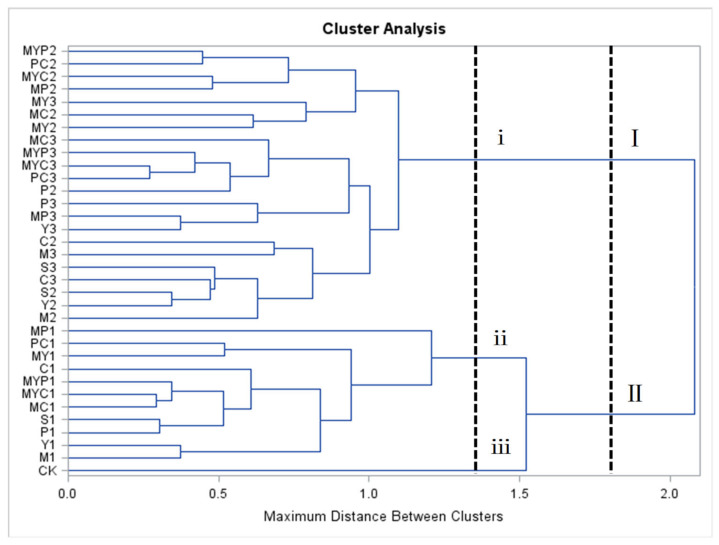
The dendrogram obtained by cluster analysis. CK, control group; M, fermented BR with *Lb. plantarum*; Y, fermented BR with *S. cerevisiae*; P, fermented BR with *R. oryzae*; C, fermented BR with *A. oryzae*; S, fermented BR with *N. sitophila*; MY, fermented BR with *Lb. plantarum* and *S. cerevisiae*; MP, fermented BR with *Lb. plantarum* and *R. oryzae*; MC, fermented BR with *Lb. plantarum* and *A. oryzae*; PC, fermented BR with *R. oryzae* and *A. oryzae*; MYP, fermented BR with *Lb. plantarum*, *S. cerevisiae,* and *R. oryzae*; MYC, fermented BR with *Lb. plantarum*, *S. cerevisiae,* and *A. oryzae*. Arabic numbers of 1, 2, and 3 were affixed after the sample labels to indicate the BR samples fermented for 12 h, 30 h, and 48 h, respectively.

**Table 1 foods-11-01560-t001:** Experimental design of control and study groups.

Groups	Label	Inoculation Proportion	BR Weight (g)	Suspension Volume (mL)					
				*Lb. plantarum*	*S. cerevisiae*	*R. oryzae*	*A. oryzae*	*N. sitophila*	Sterile Water
Control group	CK	-	300	0	0	0	0	0	90
Study group	M	*Lb. plantarum*	300	3	0	0	0	0	87
	Y	*S. cerevisiae*		0	3	0	0	0	
	P	*R. oryzae*		0	0	3	0	0	
	C	*A. oryzae*		0	0	0	3	0	
	S	*N. sitophila*		0	0	0	0	3	
	MY	*Lb. plantarum*: *S. cerevisiae* = 1:1		1.5	1.5	0	0	0	
	MP	*Lb. plantarum*: *R. oryzae* = 1:1		1.5	0	1.5	0	0	
	MC	*Lb. plantarum*: *A. oryzae* = 1:1		1.5	0	0	1.5	0	
	PC	*R. oryzae*: *A. oryzae* = 1:1		0	0	1.5	1.5	0	
	MYP	*Lb. plantarum*: *S. cerevisiae*: *R. oryzae* = 1:1:1		1.5	1.5	1.5	0	0	85.5
	MYC	*Lb. plantarum*: *S. cerevisiae*: *A. oryzae* = 1:1:1		1.5	1.5	0	1.5	0	

**Table 2 foods-11-01560-t002:** Principal component analysis results of the nutrition and sensory evaluation of the solid-state-fermented BR.

Principal Component	Eigenvalue	Difference	Proportion (%)	Cumulative (%)
*Z1*	5.2160	3.0709	52.16	52.16
*Z2*	2.1451	1.1440	21.45	73.61
*Z3*	1.0011	0.4844	10.01	83.62
*Z4*	0.5066	0.1103	5.07	88.69
*Z5*	0.4164	0.0889	4.16	92.85
*Z6*	0.3175	0.0925	3.18	96.03
*Z7*	0.2250	0.0770	2.25	98.28
*Z8*	0.1480	0.1261	1.48	99.76
*Z9*	0.0220	0.0197	0.22	99.98
*Z10*	0.0023		0.02	100.00

*Z*, principal component; *X1*, insoluble dietary fiber content; *X2*, total arabinoxylan content; *X3*, β-glucan content; *X4*, phytic acid content; *X5*, total γ-oryzanol content; *X6*, phenol content; *X7*, flavonoid content; *X8*, total antioxidant capacity; *X9*, sensory score; *X10*, L*.

**Table 3 foods-11-01560-t003:** The eigenvalues of the three principal components.

Quality Traits	*Z1*	*Z2*	*Z3*
*X1*	−0.310	−0.132	0.206
*X2*	0.166	0.557	−0.627
*X3*	0.342	−0.072	0.259
*X4*	−0.161	0.720	0.381
*X5*	0.281	−0.104	−0.134
*X6*	0.346	0.032	0.364
*X7*	0.335	0.287	0.036
*X8*	0.421	−0.022	0.182
*X9*	0.298	−0.231	−0.366
*X10*	0.395	0.020	0.184

*Z*, principal component; *X1*, insoluble dietary fiber content; *X2*, total arabinoxylan content; *X3*, β-glucan content; *X4*, phytic acid content; *X5*, total γ-oryzanol content; *X6*, phenol content; *X7*, flavonoid content; *X8*, total antioxidant capacity; *X9*, sensory score; *X10*, L*.

**Table 4 foods-11-01560-t004:** Principal component score of the solid-state-fermented BR.

Samples	Fermentation Time	*Z1*	*Z2*	*Z3*	*Z*	Ranking Order
CK	12	251.79	−17.66	35.35	1310.84	29
M1		289.47	−17.79	45.71	1517.49	23
Y1		254.17	−20.05	29.64	1312.40	28
P1		242.35	−20.31	24.77	1245.34	32
C1		230.99	−19.55	22.54	1185.46	34
S1		244.20	−18.73	27.08	1260.70	31
MY1		288.09	−21.67	44.75	1500.96	24
MP1		269.47	−16.18	31.94	1402.80	26
MC1		264.83	−19.03	36.60	1377.18	27
PC1		278.05	−21.39	37.29	1441.72	25
MYP1		241.60	−20.25	25.50	1242.28	33
MYC1		246.22	−17.58	28.01	1274.59	30
M2	30	390.90	−26.45	79.90	2062.17	11
Y2		376.20	−19.84	75.48	1995.27	14
P2		377.91	−18.45	82.68	2014.36	13
C2		414.07	−18.28	88.73	2209.39	2
S2		368.38	−22.33	68.80	1942.43	17
MY2		412.61	−17.46	84.92	2199.72	3
MP2		401.17	−17.42	86.95	2142.18	6
MC2		380.67	−20.18	74.08	2016.46	12
PC2		374.55	−23.74	68.67	1971.45	15
MYP2		348.73	−21.88	57.60	1829.72	22
MYC2		354.31	−21.52	68.39	1870.35	20
M3	48	389.85	−22.70	83.99	2068.84	10
Y3		375.21	−32.47	71.59	1959.10	16
P3		394.92	−24.05	78.97	2087.36	9
C3		409.19	−24.08	85.73	2168.51	4
S3		361.32	−26.44	71.06	1899.08	19
MY3		434.88	−13.88	97.40	2336.08	1
MP3		368.92	−29.25	70.78	1932.41	18
MC3		353.03	−25.24	67.17	1854.51	21
PC3		398.14	−22.54	88.06	2116.50	8
MYP3		402.29	−25.99	88.69	2131.38	7
MYC3		406.38	−25.36	93.78	2159.13	5

CK, control group; M, fermented BR with *Lb. plantarum*; Y, fermented BR with *S. cerevisiae*; P, fermented BR with *R. oryzae*; C, fermented BR with *A. oryzae*; S, fermented BR with *N. sitophila*; MY, fermented BR with *Lb. plantarum* and *S. cerevisiae*; MP, fermented BR with *Lb. plantarum* and *R. oryzae*; MC, fermented BR with *Lb. plantarum* and *A. oryzae*; PC, fermented BR with *R. oryzae* and *A. oryzae*; MYP, fermented BR with *Lb. plantarum*, *S. cerevisiae,* and *R. oryzae*; MYC, fermented BR with *Lb. plantarum*, *S. cerevisiae,* and *A. oryzae*. Arabic numbers of 1, 2, and 3 were affixed after the sample labels to indicate the BR samples fermented for 12 h, 30 h, and 48 h, respectively.

## Data Availability

The data that support the findings of this study are available from the corresponding author upon reasonable request.

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
