# Peer review of "Nutrition and Sensory Evaluation of Solid-State Fermented Brown Rice Based on Cluster and Principal Component Analysis"

_foods, 2022, doi:10.3390/foods11111560_

Round 1
Reviewer 1 Report
General comments:
This manuscript is well-prepared with excellent clarity. The tables and figures are carefully designed and concisely interpreted and discussed. I enjoy the reading.
Specific comments:
Line 20: need full name for CK
Line 58-61: is this your own thought? or a citation is required.
Line 60: “functional foods (enzyme)”: do you mean enzyme is a functional food(s)?!
Line 62-70: what kinds of food applications were in your mind when you designed these experiments/variables?! A microbial strain that is good for yoghurt could be bad for the winery, please make your target application clear here.
Line 101: five microbial strains, while only “four media”, should be matched?
Line 107: “at 4C until used” is not correct. It should not be stored too long. Please provide the best range of time.
Line 101: Where is Fig. S1? I cannot find it.
Line 113-118: how was the mixing done? How was the milling done after freeze-drying?
Line 124: citation and AOAC method number are needed.
Line 128: “vortexed for 10s at 3500 x g”: confusing! Vortex doesn’t have 3500 x g, but it was centrifuged, 10 s is so short!
Line 129: “1 mL aliquots 128 were transferred to new Pyrex tubes”: what this portion was used for? Cannot find it anywhere in your full text.
Line 133: typo. move period to the right position.
Line 135 & 138: citation is definitely needed.
Line 141: Instrument parameters for ultrxonic are needed for repeating your experiment with your peers.
Line 143: what is “contentated”? is it concentrated?!
Line 146: anto should be auto.
Line 154: “was kept for further use”: where is the part in your full text? Cannot find it.
Line 166: citation is needed for this method.
Line 176: citation is needed for this method.
Line 191-192: criteria for scoring are needed, or citation can be used.
Figure 1 & 2: Since fermentation time (12 h, 30 h, 48 h) is a continuous variable, line+symbol+error bar is recommended instead of a bar chart.
Figure 2B: where are error bars and statistical letters?
Line 315: “0.5357”? Figure 3 seems to show the DBC>1.5 for two clusters and DBC = 1.5 to get 3 clusters.
Line 326: citation is needed at the end of this paragraph.
Table 2: PCA has a proper name for this table, is this a table of eigenvalues?
Line 378: “sustainable” has no evidence from your research data.
Author Response
Specific comments:
#Reviewer 1:
(1) Line 20: need full name for CK.
Response: We have modified the “CK” to “the control group” according to the reviewer’s suggestion; please see line 21 in the revised manuscript for details.
(2) Line 58-61: is this your own thought? or a citation is required.
Response: We have supplemented a citation according to the reviewer’s suggestion; please see line 69 in the revised manuscript for details.
(3) Line 60: “functional foods (enzyme)”: do you mean enzyme is a functional food(s)?!
Response: We have corrected “enzyme” into “probiotic complex” according to the reviewer’s suggestion; please see line 68 in the revised manuscript for details.
(4) Line 62-70: what kinds of food applications were in your mind when you designed these experiments/variables?! A microbial strain that is good for yoghurt could be bad for the winery, please make your target application clear here.
Response: We have supplemented the target application according to the reviewer’s suggestion; please see line 73-77 in the revised manuscript for details.
(5) Line 101: five microbial strains, while only “four media”, should be matched?
Response: We have supplemented the statement about this according to the reviewer’s suggestion. Please see line 98 in the revised manuscript for details.
(6) Line 107: “at 4C until used” is not correct. It should not be stored too long. Please provide the best range of time.
Response: We have supplemented the best range of time according to the reviewer’s suggestion. Please see line 135-136 in the revised manuscript for details.
(7) Line 101: Where is Fig. S1? I cannot find it.
Response: We have added Supplementary Figure 1 according to the reviewer’s suggestion. Please see line 140 in the revised manuscript and the revised supplementary figure.
(8) Line 113-118: how was the mixing done? How was the milling done after freeze-drying?
Response: We have detailed the description of the mixing operations according to the reviewer’s suggestion. Please see line 149-151 and 153-156 in the revised manuscript for details.
(9) Line 124: citation and AOAC method number are needed.
Response: We have supplemented the AOAC method number and briefly described the method according to the reviewer’s suggestion. Please see line 162-169 in the revised manuscript for details.
(10) Line 128: “vortexed for 10s at 3500 x g”: confusing! Vortex doesn’t have 3500 x g, but it was centrifuged, 10 s is so short!
Response: We have corrected “vortexed for 10 s at 3500 x g” into “vortexed for 10 s”. Please see line 173 in the revised manuscript for details.
(11) Line 129: “1 mL aliquots were transferred to new Pyrex tubes”: what this portion was used for? Cannot find it anywhere in your full text.
Response: We have modified this part according to the reviewer’s suggestion. Please see lines 174-178 in the revised manuscript for details.
(12) Line 133: typo. move period to the right position.
Response: We have move the period to the right position according to the reviewer’s suggestion. Please see line 183 in the revised manuscript for details.
(13) Line 135 & 138: citation is definitely needed.
Response: We have supplemented the citation and the briefly operation according to the reviewer’s suggestion. Please see line 184-198 and 201-210 in the revised manuscript for details.
(14) Line 141: Instrument parameters for ultrxonic are needed for repeating your experiment with your peers.
Response: We have corrected “ultrxonic” into “ultrasonic” and added the instrument parameters for ultrasonic according to the reviewer’s suggestion. Please see line 214-215 in the revised manuscript for details.
(15) Line 143: what is “contentated”? is it concentrated?!
Response: We have corrected “contentated” into “concentrated” in the revised manuscript. Please see line 217 for details.
(16) Line 146: anto should be auto.
Response: We have corrected “anto” to “auto” in the revised manuscript. Please see line 220 for details.
(17) Line 154: “was kept for further use”: where is the part in your full text? Cannot find it.
Response: We have deleted “The obtained precipitates were kept for further use” in the revised manuscript. Please see line 233 for details.
(18) Line 166: citation is needed for this method.
Response: We have supplemented the citation of the determination of flavonoid content according to the reviewer’s suggestion. Please see line 245-246 in the revised manuscript for details.
(19) Line 176: citation is needed for this method.
Response: We have supplemented the citation according to the reviewer’s suggestion. Please see line 256-260 in the revised manuscript for details.
(20) Line 191-192: criteria for scoring are needed, or citation can be used.
Response: We have supplemented the criteria for scoring according to the reviewer’s suggestion. Please see the Supplementary Table 1 in the revised supplementary file and line 274 in the revised manuscript for details.
(21) Figure 1 & 2: Since fermentation time (12 h, 30 h, 48 h) is a continuous variable, line+symbol+error bar is recommended instead of a bar chart.
Response: Appreciate for your kind suggestion. Because of the large amount of data, bar chart was more preferable to clearly present the information.
(22) Figure 2B: where are error bars and statistical letters?
Response: We have supplemented the error bars and statistical letters in Figure 2B according to the reviewer’s suggestion. Please see Figure 2B in the revised manuscript for details.
(23) Line 315: “0.5357”? Figure 3 seems to show the DBC>1.5 for two clusters and DBC = 1.5 to get 3 clusters.
Response: We have corrected the value in the revised manuscript. Please see line 432 and 434 for details.
(24) Line 326: citation is needed at the end of this paragraph.
Response: We have supplemented the citation according to the reviewer’s suggestion. Please see line 443 and 634-636 in the revised manuscript for details.
(25) Table 2: PCA has a proper name for this table, is this a table of eigenvalues?
Response: We have supplemented Table 2 to show the principal component analysis results, and changed the caption of Table 3 into “The eigenvalues of the three principal components” according to the reviewer’s suggestion. Please see line 472-478 in the revised manuscript for details.
(26) Line 378: “sustainable” has no evidence from your research data.
Response: We have deleted “sustainable” according to the reviewer’s suggestion. Please see line 518 in the revised manuscript for details.

Reviewer 2 Report
The work is interesting and well written. The experiment was well planned and the results are relevant. I only propose some text corrections, marked in yellow in the pdf I attach.

Author Response
#Reviewer 2:
(1) Cluster analysis.
Response: We have changed “cluster” into “cluster analysis” according to the reviewer’s suggestion. Please see the revised manuscript for details.
(2) Please define CK.
Response: We have modified the “CK” to “the control group” according to the reviewer’s suggestion. Please see line 18-19 the revised manuscript for details.
(3) Lactiplantibacillus plantarum, formely Lactobacillus plantarum, Lactiplantibacillus plantarum (Lb. plantarum).
Response: We have changed “Lactobacillus plantarum (L. plantarum)” to “Lactiplantibacillus plantarum (Lb. plantarum)” according to the reviewer’s suggestion. Please see all the revised manuscript for details.
(4) Define the method.
Response: We have supplemented the AOAC method according to the reviewer’s suggestion. Please see line 162-169 in the revised manuscript for details.
(5) Add more Information about the Kit.
Response: We have added more information about the Mixed-linkage β-glucan assay kit according to the reviewer’s suggestion. Please see line 184-198 in the revised manuscript for details.
(6) Add more information about the kit.
Response: We have added more information about the Phytic acid (Phytate)/total phosphorus assay kit according to the reviewer’s suggestion. Please see line 202-210 in the revised manuscript for details.

Reviewer 3 Report
The manuscript is written with clear understanding of the project addressed. However, there are major concerns that need to be addressed to enhance the quality of the manuscript. My specific comments are as follows:
Abstract:
Elaborate more on the methods used in this study.
Elaborate more on the specific finding
Introduction:
Page1Line44: “In particular, lots of studies have reported that germination…” Elaborate on the findings of these studies
Based on your objectives, please compare how your study is different from those that have already been published
Materials and Methods:
P2L90: “The proximate compositions of BR (dry basis) were 2.12 g/100 g fat, 1.35 g/100g ash, 8.60 g/100g protein, 77.39 g/100g starch, and 3.65 g/100g dietary fibre.” Explain the methods on how do you conducted proximate analysis.
How many samples of brown rice used in this study?
S2.2.3: What are the acronym for control group and study group represented for? (CK, M, Y, P, etc)
S2.3.1: “Soluble (SDF) and insoluble dietary fiber (IDF) contents of the BR samples were determined according to the AOAC methods.” Explain briefly the methods
S2.3.5: “The TOC was calculated from the peak area obtained by the HPLC analysis.” Explain briefly the method
S2.4.2: involved- involved (check grammar error for entire manuscript)
Results and Discussion:
Explain briefly type of samples for Fig.1. Put the figure number such as 1A, 1B, 1C, and so on to respective figures
How about the results for a* and b* (color analysis)?
S3.2: “After SSF, the sensory score increased from 59.82 to 74.22, indicating the obviously improvement of the comprehensive sensory characteristics.” For which sample? How about other samples?
Table 3: Add the total variances to show the differentiation between each fermentation time and samples.
Based on the obtained results., instead of mentioning the results, discuss the relevant findings of getting those values/scores.
Conclusions:
Add on recommendation for future studies.
Which one is better? CA or PCA in terms of giving the best results
General comments:
Please check the reference styles and grammar of the manuscript.
Author Response
#Reviewer 3:
The manuscript is written with clear understanding of the project addressed. However, there are major concerns that need to be addressed to enhance the quality of the manuscript. My specific comments are as follows:
(1) Abstract: Elaborate more on the methods used in this study. Elaborate more on the specific finding.
Response: We have elaborate more on the methods and specific finding in the abstract according to the reviewer’s suggestion. Please see line 13-24 in the revised manuscript for details.
(2) Introduction: Page1Line44: “In particular, lots of studies have reported that germination…” Elaborate on the findings of these studies.
Response: We have summarized and elaborate the finding of these studies according to the reviewer’s suggestion. Please see line 49-52 in the revised manuscript for details.
(3) Based on your objectives, please compare how your study is different from those that have already been published.
Response: We have supplemented the statements according to the reviewer’s suggestion. Please see line 69-77 in the revised manuscript for details.
(4) Materials and Methods:
P2L90: “The proximate compositions of BR (dry basis) were 2.12 g/100 g fat, 1.35 g/100g ash, 8.60 g/100g protein, 77.39 g/100g starch, and 3.65 g/100g dietary fibre.” Explain the methods on how do you conducted proximate analysis.
Response: We have supplemented the determination method of the proximate composition of BR according to the reviewer’s suggestion. Please see section 2.2 in the revised manuscript for details.
(5) How many samples of brown rice used in this study?
Response: The same brown rice cultivar of Suijing No. 18 (kindly provided by Wuchang Jin He Rice Industry Co., Ltd., Heilongjiang, China) was used in this study. Please see section 2.1 (line 89-90) in the revised manuscript for details.
(6) S2.2.3: What are the acronym for control group and study group represented for? (CK, M, Y, P, etc)
Response: We have modified Table 1 and added more descriptions of the labeling rules of the sample according to the reviewer’s suggestion. Please see Table 1 and line 144-147 in the revised manuscript for details.
(7) S2.3.1: “Soluble (SDF) and insoluble dietary fiber (IDF) contents of the BR samples were determined according to the AOAC methods.” Explain briefly the methods
Response: We have supplemented the briefly explanation of the method according to the reviewer’s suggestion. Please see line 162-169 in the revised manuscript for details.
(8) S2.3.5: “The TOC was calculated from the peak area obtained by the HPLC analysis.” Explain briefly the method.
Response: We have briefly explained the method according to the reviewer’s suggestion. Please see line 223-227 in the revised manuscript for details.
(9) S2.4.2: involved- involved (check grammar error for entire manuscript).
Response: We have corrected “invovled” into “involved”, and modified grammar errors for entire manuscript. Please see line 276 in the revised manuscript for details.
(10) Results and Discussion:
Explain briefly type of samples for Fig.1. Put the figure number such as 1A, 1B, 1C, and so on to respective figures.
Response: We have added briefly explain of the type of samples for Figure 1 and put the figure number to respective figures according to the reviewer’s suggestion. Please see Figure 1 and Figure 2 in the revised manuscript for details.
(11) How about the results for a* and b* (color analysis)?
Response: We have supplemented the statements about the results of a* and b* according to the reviewer’s suggestion. Please see line 386-387 in the revised manuscript for details.
(12) S3.2: “After SSF, the sensory score increased from 59.82 to 74.22, indicating the obviously improvement of the comprehensive sensory characteristics.” For which sample? How about other samples?
Response: We have supplemented the explanation according to the reviewer’s suggestion. Please see line 413-414 in the revised manuscript for details.
(13) Table 3: Add the total variances to show the differentiation between each fermentation time and samples.
Response: Appreciate for your comments. However, Table 3 has been changed to Table 4, which was obtained according to the principal components and its proportion (as shown in Table 2 and 3) without variances.
(14) Based on the obtained results, instead of mentioning the results, discuss the relevant findings of getting those values/scores.
Response: We have modified the manuscript according to the reviewer’s suggestion. Please see line 485-486 and 493-495 in the revised manuscript for details.
(15) Conclusions: Add on recommendation for future studies.
Response: We have added the recommendation for future studies according to the reviewer’s suggestion. Please see line 520-523 in the revised manuscript for details.
(16) General comments: Please check the reference styles and grammar of the manuscript.
Response: We have checked the reference styles and grammar of the manuscript according to the reviewer’s suggestion. Please see the revised manuscript for details.
